# Chemoselective Synthesis and Anti-Kinetoplastidal Properties of 2,6-Diaryl-4*H*-tetrahydro-thiopyran-4-one *S*-Oxides: Their Interplay in a Cascade of Redox Reactions from Diarylideneacetones

**DOI:** 10.3390/molecules29071620

**Published:** 2024-04-04

**Authors:** Thibault Gendron, Don Antoine Lanfranchi, Nicole I. Wenzel, Hripsimée Kessedjian, Beate Jannack, Louis Maes, Sandrine Cojean, Thomas J. J. Müller, Philippe M. Loiseau, Elisabeth Davioud-Charvet

**Affiliations:** 1UMR7042 Université de Strasbourg–CNRS–UHA, Laboratoire d’Innovation Moléculaire et Applications (LIMA), Team Bio(IN)organic and Medicinal Chemistry, European School of Chemistry, Polymers and Materials (ECPM), 25 Rue Becquerel, F-67087 Strasbourg, France; t.gendron@uliege.be (T.G.); don.antoine.lanfranchi@gmail.com (D.A.L.); hripsimee.a.kessedjian@gsk.com (H.K.); 2Bioorganic & Medicinal Chemistry Laboratory, Biochemie-Zentrum, Heidelberg University, Im Neuenheimer Feld 504, D-69120 Heidelberg, Germany; n.wenzel@gmx.net (N.I.W.);; 3Laboratory of Microbiology, Parasitology and Hygiene (LMPH), Faculty of Pharmaceutical, Biomedical and Veterinary Sciences, University of Antwerp, Universiteitsplein 1, B-2610 Antwerp, Belgium; louis.maes@uantwerpen.be; 4Antiparasitic Chemotherapy, Faculty of Pharmacy, BioCIS, UMR 8076 Université Paris-Saclay-CNRS 17, Rue des Sciences, F-91400 Orsay, France; sandrine.cojean@universite-paris-saclay.fr (S.C.); philippe.loiseau@universite-paris-saclay.fr (P.M.L.); 5Institut für Organische Chemie und Makromolekulare Chemie, Mathematisch-Naturwissenschaftliche FakultätFakultät, Heinrich-Heine-Universität Düsseldorf, Universitätsstraße 1, D-40225 Düsseldorf, Germany; thomasjj.mueller@uni-duesseldorf.de

**Keywords:** anti-kinetoplastid, drug metabolites, Michael acceptor, redox, sulfide, sulfone, sulfoxide

## Abstract

2,6-Diaryl-4*H*-tetrahydro-thiopyran-4-ones and corresponding sulfoxide and sulfone derivatives were designed to lower the major toxicity of their parent anti-kinetoplatidal diarylideneacetones through a prodrug effect. Novel diastereoselective methodologies were developed and generalized from diarylideneacetones and 2,6-diaryl-4*H*-tetrahydro-thiopyran-4-ones to allow the introduction of a wide substitution profile and to prepare the related *S*-oxides. The in vitro biological activity and selectivity of diarylideneacetones, 2,6-diaryl-4*H*-tetrahydro-thiopyran-4-ones, and their *S*-sulfoxide and sulfone metabolites were evaluated against *Trypanosoma brucei brucei*, *Trypanosoma cruzi*, and various *Leishmania* species in comparison with their cytotoxicity against human fibroblasts *h*MRC-5. The data revealed that the sulfides, sulfoxides, and sulfones, in which the Michael acceptor sites are temporarily masked, are less toxic against mammal cells while the anti-trypanosomal potency was maintained against *T. b. brucei*, *T. cruzi*, *L. infantum*, and *L. donovani*, thus confirming the validity of the prodrug strategy. The mechanism of action is proposed to be due to the involvement of diarylideneacetones in cascades of redox reactions involving the trypanothione system. After Michael addition of the dithiol to the double bonds, resulting in an elongated polymer, the latter—upon *S*-oxidation, followed by *syn-*eliminations—fragments, under continuous release of reactive oxygen species and sulfenic/sulfonic species, causing the death of the trypanosomal parasites in the micromolar or submicromolar range with high selectivity indexes.

## 1. Introduction

Three distinct kinetoplastid parasites cause human diseases: *Trypanosoma brucei* (human African trypanosomiasis (HAT) or sleeping sickness), *T. cruzi* (Chagas’ disease), and *Leishmania* spp. (leishmaniasis), which are considered among the top priorities by the World Health Organization and recognized as neglected tropical diseases (NTDs) [1]. Despite the advances in the development of drug therapies and vector control, new therapeutics are required for combatting emerging drug resistance.

Kinetoplastid parasites possess a unique redox metabolism based on the specific dithiol trypanothione (T(SH)_2_) to maintain the essential redox equilibrium in the human host [2]. The central and exclusive player is the NADPH-dependent flavoenzyme trypanothione reductase (TR). The known sensitivity of these parasites toward oxidative stress and the absence of the enzyme in the mammalian host render the T(SH)_2_/TR system an attractive target for the development of novel anti-trypanosomal drugs [2]. Only a few drugs are currently available and unfortunately, most have limited efficacy, especially during the chronic phase of the diseases, in addition to suboptimal pharmacokinetic properties. In addition, relapses and severe side effects may lead to the interruption of treatment. Interestingly, many of these old drugs do target the T(SH)_2_/TR system, such as benznidazole/nifurtimox for Chagas’ disease, arsenicals for sleeping sickness, or antimonials for leishmaniasis [3]. However, safer and more efficacious drugs are needed due to multidrug resistance.

Unsaturated Mannich bases [4,5,6,7] and curcuminoids, including curcumin or related diarylideneacetones (DAAs, namely, for the 1,5-diaryl penta-1,4-dien-3-ones) [8,9,10,11], were shown to kill kinetoplastid parasites through T(SH)_2_ depletion and T(SH)_2_/TR alkylation or by inducing oxidative stress (ROS). In spite of these numerous applications, only a few studies have reported the use of curcuminoids or DAA derivatives for antiparasitic purposes [12,13,14,15,16]. As part of our preliminary research effort to develop novel antiparasitic drugs, we have built a library of 27 symmetrical DAAs **1a**–**1za** and 19 unsymmetrical DAAs **1′a**–**1′s** with various substitution patterns on the aromatic rings (Appendix A). Unlike Mannich bases or divinylketones [4,5] these products were stable both in solid state and in solution and were therefore screened in parasitic in vitro assays for the determination of the most active substitution pattern. Potency results were excellent with nanomolar range IC_50_ values, but the anti-trypanosomal activity of selected representatives was clearly associated with high toxicity (in the micromolar range) toward human *h*MRC-5 cells (Table 1) [17].

The observed cytotoxicity is likely due to the high reactivity of the electrophilic centers of DAAs with human glutathione or parasitic trypanothione. Indeed, we demonstrated the formation of mono- and double adducts of glutathione (GSH) or oligomers of trypanothione adducts on DAA’s electrophilic centers of DAAs **1r** (Figure 1 and Appendix A) and DAA **1s**, in agreement with reported studies on the mechanism of action (MoA) of a curcuminoid series [15,17]. Indeed, the anti-trypanosomal activity was attributed to the formation of trypanothione adducts as a result of Michael addition between the mono-enone group of curcumin analogs and trypanothione, depleting the parasite of essential thiols and leading to parasite death [15,17].

To temporarily mask both Michael addition sites of DAAs **1**, we designed a series of 2,6-diaryl-4*H*-tetrahydrothiopyran-4-ones (2,6-DA-4-THTPs) **2**–**2′** (Figure 2 and Appendix A) as prodrugs of DAAs and developed a diastereoselective synthesis of this scaffold [18,19]. Interestingly, 2,6-DA-4-THTPs were also considered as potential antimalarial agents [20,21].

Sulfoxides or sulfones are known metabolites of sulfides in vivo [22]. The recently launched fexinidazole to cure HAT has been shown to generate key *S*-oxide metabolites against *T. brucei* [23,24]. It is well documented that *S*-oxides can undergo β-elimination, resulting in the formation of a double bond through pericyclic [25,26], ionic, photolytic, or radical mechanisms [27,28,29,30,31] (Figure 3).

Once integrated into organic compounds, the sulfur atom, with its several degrees of oxidation, will be submitted to many and varied metabolic fates in living cells through different mechanisms [32,33,34,35,36,37,38,39,40,41]. In the field of organic chemistry, the synthesis and the reactivity of sulfur-containing compounds have also been extensively exploited to generate double bonds, as described in the literature [32,33,34,35,36,37,38,39,40,41]. In particular, it is well known that sulfoxide and sulfone can undergo β-elimination, thus resulting in the formation of a double bond (Figure 3).

Considering the various *S*-based drug metabolism pathways (Figure 3), we anticipated that the 2,6-DA-4-THTPs could act as precursors of active DAA through the formation of *S*-oxide metabolites in vivo (Figure 2 and Figure 4).

Under physiological conditions, ionic and radical mechanisms can reasonably be considered as potential metabolic pathways for 2,6-DA-4-THTP. Indeed, the host immune response is responsible for highly oxidative stress near parasites, implying a considerable amount of ROS and free radicals. Also, basic catalysis is possible in any enzymatic pocket or specific biological compartments. Hypothesizing that the *S*-oxides of 2,6-DA-4-THTP could also be active prodrugs of interest, we sought to synthesize a small library of sulfoxide and sulfone analogs and test them for their antiparasitic activity and toxicity (Figure 2). To the best of our knowledge, the synthesis of 2,6-DA-4-THTP *S*-oxides has never been thoroughly described, nor have the stereochemistry aspects been discussed in detail. Here, we disclose efficient methods to synthesize and determine the configuration of 2,6-DA-4-THTP *S*-oxides.

Regarding the in vitro evaluation, all starting symmetrical DAAs **1a**–**1za**, unsymmetrical DAAs **1a′**–**1s′**, 2,6-DA-4-THTP derivatives **2** and **2′**, and their related *S*-oxides **3**–**4** and **3′**–**4′** were first tested in vitro against *T. b. brucei* trypomastigotes, intracellular *T. cruzi* amastigotes, and axenic or intracellular *L. infantum* or *L. donovani* amastigotes in primary screening assays. We further propose a reaction mechanism for 2,6-DA-4-THTPs as prodrugs of starting DAAs, being metabolized in a pathway where, upon *S*-oxidation, DAAs enter a cascade of redox reactions interplaying with the T(SH)_2_/TR system.

## 2. Results

### 2.1. Optimized Synthesis of Starting Symmetrical and Unsymmetrical Diarylideneacetones **1** and **1′** and Symmetrical (±)-trans and cis-2,6-Diaryl-4H-tetrahydrothiopyran-4-ones **2** and **2′**

A library of 27 symmetrical DAAs **1** and 19 unsymmetrical DAAs **1′** was produced through improved and versatile methodologies to be used per se in anti-parasitic drug screening assays (Appendix A). Then, the synthesized 27 symmetrical DAAs **1** (**1a**–**1za**) were used as starting materials for the diastereoselective synthesis of 17 (±)-*trans*- and 13 *cis*-2,6-DA-4-THTP **2** and **2′** (Appendix A) in the conditions reported earlier [19] and in optimized conditions, as described here. In the reported work [19], the establishment of reliable criteria for the unambiguous spectroscopic determination of the diastereoisomeric configuration of 2,6-DA-4-THTPs was an important stage prior to follow-up synthetic work. Further to the identification of key parameters liable to influence both the yield and the diastereoisomeric ratio, we followed an optimization process similar to the one used for the synthesis of unsymmetrical DAAs, which resulted in the identification of three sets of conditions for the selective synthesis of (±)-*trans* **2** and *cis*-2,6-DA-4-THTP **2′** isomers. These optimized reaction conditions were applied to a wide panel of substitution patterns with moderate to good diastereoisomeric excesses and good to excellent yields (Appendix A).

### 2.2. Synthesis of 2,6-Diaryl-4H-tetrahydrothiopyran-4-one S-Oxides

In our approach to gather as much information as possible about the 2,6-DA-4-THTP series, the reactivity of the sulfur toward several oxidants was studied. The 2,6-DA-4-THTP *S*-oxides have not been described extensively in the literature, and most of the described protocols lack efficiency and chemoselectivity [20]. Moreover, toxic hazardous reagents were used in these procedures. Here, we describe reliable mild methods to selectively synthesize the corresponding sulfoxides and sulfones of (±)-*trans*- and *cis*-2,6-DA-4-THTP isomers. Examples are given with the anisyl substitution (starting materials **2d** and **2′d**), and the described procedures were successfully generalized to other substitutions, including heterocyclic ones such as those prepared from starting materials **2w** and **2′w**.

#### 2.2.1. Synthesis of 2,6-DA-4-THTP Sulfoxide Derivatives **3**–**3′**

To our knowledge, the synthesis of 2,6-DA-4-THTP sulfoxides derivatives has been described in only three articles, which were limited to the oxidation of the *cis* isomer [42,43,44]. In each case, wet bromine was used to oxidize the sulfur, with a variable yield of up to 83%. When this procedure was used on our substrates, lower yields were observed with sensitive substitutions, especially with anisyl-, *N*-phenylacetamide-, and furyl-substituted 2,6-DA-4-THTPs. The analysis of the crude revealed that the starting material was entirely converted to the desired product, while a non-negligible amount of degradation products was observed. This may be explained by the high reactivity of bromine, associated with a lack of chemoselectivity that is reinforced by the relatively high reaction temperature (0 °C). Facing these difficulties, we investigated a new procedure for the synthesis of these sulfoxides, although numerous protocols are described in the literature for the oxidation of sulfur to sulfoxide [45]. Among them, chemoselective, mild, or catalyzed protocols were selected and evaluated toward our substrates. With the well-known conditions described by Brunel and Kagan [46], most of the starting material was recovered, and a negligible amount of the desired product was detected (~25 mol%). With the aim to improve this result, modified conditions of Kagan’s protocol were used [47]. Unfortunately, these modifications were detrimental to obtaining the desired product as the sulfoxide was detected with a lower (15%) yield. Considering that Kagan’s conditions might be not sufficient to oxidize the 2,6-DA-4-THTP core, the oxidative system was shifted to a stronger one with the use of one equivalent of peracetic acid alone in dichloromethane. This time, the desired sulfoxide was isolated in 50% yield, with a large amount of sulfone resulting from the overoxidation. Considering these disappointing results obtained with usual oxidative conditions, we shifted to the use of more specific reagents such as oxaziridines. Indeed, Davis and coworkers already demonstrated the exceptional ability of these species to transfer an amine group or an oxygen atom to various nucleophiles [48]. For the oxidation of sulfur to sulfoxide, the so-called Davis’s oxaziridine proved its efficiency [49]. The use of this specific reagent allowed the synthesis of the desired (±)-*trans*- and *cis*-2,6-DA-4-THTP sulfoxide isomers **3d** and **3′d/3″d** in 82% and 86% yields, respectively (Figure 5). It is of interest to note that the reaction is completely chemoselective without overoxidation and proceeds smoothly in dichloromethane in normally less than three hours. As an example of this chemoselectivity, this method was tested with 2,6-[2-pyridyl-substituted]-4*H*-tetrahydrothiopyran-4-one **2w** and **2′w**. No oxidation was observed on the nitrogen, leading to the desired (±)-*trans*- and *cis*-sulfoxide isomers **3w** and **3′w** in 73% and 80% yields, respectively (Figure 5). The reaction was carried out successively at −78 °C, −20 °C, and 0 °C without significant changes in yield, purity, or reaction time. Thus, this procedure is highly adaptable to the sensitivity of the substrate and the product.

From a stereochemistry point of view, the oxidation of the *cis* sulfide should give two different diastereoisomers, depending on the position of the oxygen. In one case, it is in an equatorial position (hereafter quoted as *anti*-*cis* sulfoxide isomer), whereas in the other case, it is in an axial position (hereafter quoted as *syn*-*cis* sulfoxide isomer) (Figure 1).

The most stable configuration of the sulfoxide for this kind of heterocycle has been discussed in several studies [42,50,51]. According to these works, the axial position is surprisingly more stable than the equatorial one; the *syn*-*cis* is therefore expected to be the major isomer, if not the unique one. With the oxaziridine-based oxidation of sulfide **2′b**, two products were isolated. NMR and LCMS analyses were fine for sulfoxide for both products. Further to previous studies [42,43,44], the NMR spectra of the major isomer agree with an axial configuration of the sulfoxide.

One publication briefly described NMR data for sulfoxide derivatives of 2,6-DA-4-THTP [42], in which Klein and Stollar used the significant deshielding of the axial proton in the β-position to the sulfoxide as proof of the axial positioning of the oxygen atom. This argument was supported by another study on similar sulfoxide-containing heterocycles [50]. Basing our investigations on these weak NMR data, we considered the variation of chemical shifts for relevant protons of sulfoxide isomers compared to the same protons in sulfide (Table 2).

Regarding the axial proton in the α-position to the sulfoxide (H_X(ax)_), variations of the chemical shift are similar in both oxidized isomers **3′d** and **3″d**, with a shielding of about δ 0.20 ppm. On the other hand, protons in the β-position to the sulfoxide, and especially, axial ones, are significantly influenced by the presence of the oxygen. Product **3′d** is the most affected as the deshielding of H_A(ax)_ reaches δ 0.72 ppm. In the meantime, equatorial proton H_B(eq)_ is shielded by about δ 0.26 ppm. Isomer **3″d** is significantly less affected as almost no variation is observed for the equatorial proton, whereas the axial is shifted to a lower field from δ 0.23 ppm.

According to these observations and the aforementioned studies, it could be concluded that the major isomer **3″d** has its oxygen in an axial position, thus being the *syn*-*cis*-sulfoxide isomer. This conclusion is in contradiction to that of Davis and coworkers, who concluded on an equatorial positioning of the oxygen when sulfoxides are obtained further to oxidation with Davis’s oxaziridine **6** [48]. Without any other reliable references, it remains challenging to attribute configurations of each isolated isomer, and therefore, we did not form any conclusions regarding this point.

To summarize this part of our work, we describe a new method for the efficient oxidation of 2,6-DA-4-THTPs to their relative sulfoxides. The use of Davis’s oxaziridine **6** as an oxidative agent has several advantages. The high chemoselectivity and tolerance of this reagent allow selective oxidation of the sulfur without side reactions. A wide range of temperature—from −78 °C to 0 °C—is suitable for this reaction, thus being compatible with thermo-sensitive molecules. Finally, Davis’s oxaziridine is easy to prepare and handle, stable over months, and, above all, less toxic and hazardous than bromine.

#### 2.2.2. Synthesis of 2,6-DA-4-THTP Sulfone Derivatives **4**–**4′**

The synthesis of 2,6-DA-4-THTPs sulfone derivatives may be considered easier than the preparation of sulfoxides as there is no more chirality on the sulfur atom and an oxidizing agent can be used in excess. In addition, this oxidation step was previously reported in several publications [52,53]. Reactions were performed by mixing starting 2,6-DA-4-THTP with peracetic acid in ethyl acetate or dichloromethane, overnight and at high molarity. With the constant aim to develop less drastic procedures, we looked for a milder or catalytic protocol for the synthesis of 2,6-DA-4-THTP sulfone derivatives. Interestingly, we observed that the direct mild oxidation of sulfides **2–2′** to sulfones **4–4′** remained difficult. Ammonium molybdate or sodium tungstate are known to promote the mild oxidation of sulfide to sulfone by hydrogen peroxide [54,55]. Both systems were tested but neither allowed the synthesis of the desired product. Likewise, the use of diluted mCPBA overnight at room temperature only gave a 32% yield. One solution to circumvent this difficulty linked to the formation of overoxidized sulfides is to first synthesize the sulfoxide and then oxidize it in sulfone. With starting 2,6-DA-4-THTP sulfoxides **3–3′**, this approach was successful as the sulfones **4** and **4′** were finally obtained in excellent yields by smooth and slow oxidation of parent sulfoxides **3** and **3′** with mCPBA in dichloromethane at low temperature (Figure 6). The four synthesized sulfones were submitted for biological evaluation.

### 2.3. Anti-Kinetoplastid Activities and Cytotoxicity

#### 2.3.1. Primary Evaluation of Diarylideneacetones and 2,6-Diaryltetrahydrothiopyran-4-ones

A library of 46 diverse DAAs was synthesized as starting materials for the preparation of 2,6-DA-4-THTPs **2** and **2′** and their *S*-oxides, including the sulfoxides **3** and **3′** and sulfones **4** and **4′**. All compounds were evaluated for their anti-kinetoplastid activity and cytotoxicity in primary screenings in vitro: the DAAs (Appendix A), 2,6-DA-4-THTP (Appendix A), sulfoxides (Appendix A), and sulfones (Appendix A).

#### 2.3.2. Primary Evaluation of 2,6-Diaryltetrahydrothiopyran-4-one *S*-Oxides

As they are prone to easily undergo β-elimination, 2,6-DA-4-THTP sulfoxide and sulfone derivatives are expected to allow a rapid regeneration of the active DAA and thus, the anti-kinetoplastid activity. This might compensate for the lack of activity observed in most of the parent sulfides **2** and **2′**. In order to check the validity of this hypothesis, five sulfoxides and four sulfones were submitted for biological evaluation (Figure 2).

To support the proof-of-concept, our preliminary studies were restricted to two different substitution patterns. Based on the results obtained in the DAA series, we chose the anisyl- and 2-pyridyl-substituents as representative examples for the evaluation of putative *S*-oxide metabolites. For the sake of the comparison, in Appendix A, bracketed values in blue refer to the IC_50_/CC_50_ values of the parent sulfides (2,6-DA-4-THTPs), and those in red wine refer to the IC_50_/CC_50_ values of the parent DAAs. All parent DAAs, including the unsymmetrical ones [56], and 2,6-DA-4-THTPs [18,19] were freshly synthesized before the biological evaluation for comparison with the 2,6-DA-4-THTP *S*-oxides.

At first glance, the anti-trypanosomal activity of sulfoxide derivatives was restored compared to the parent sulfides (Table 3 and Table 4). Although anisyl- and 2-pyridyl-substituted sulfides were displaying IC_50_ values higher than 26 µM, their relative sulfoxides had IC_50_ values in the low micromolar range, but with a concomitant increase in cytotoxicity. The pyridyl-substituted sulfoxides showed cytotoxicity around 8 µM (Table 3), which is more limited compared to the original DAA with a selectivity index of about 10. Interestingly, products **3w** and **3′w** were the first metabolites that showed a good potency toward both *Trypanosoma* and *Leishmania* while having an acceptable selectivity index. Similarly, sulfoxides bearing the anisyl substituent were more active than their relative sulfides (Table 4). The toxicity of the oxidized form was slightly higher than the parent DAA, resulting in a slight decrease in the selectivity index.

From a stereochemical point of view, it is interesting to note that the two separated *cis*-sulfoxides **3″d** and **3′d** behaved differently. Although **3′d** was inactive against both *T. cruzi* and *L. infantum*, its diastereoisomer counterpart **3″d** showed moderate potency against both. Furthermore, the toxicity of sulfoxide **3″d** was higher than **3′d**. These results tend to prove that the sulfoxide stereochemistry might affect biological activities. However, as the exact configuration of both sulfoxides remains unknown, and considering the fact that we had only one example illustrating such an effect, it is difficult to form conclusions based on this observation.

The results for the series of sulfone derivatives (Table 3 and Table 4) were very similar to the ones of the sulfoxide derivatives. 2-Pyridyl-substituted sulfones **4w** and **4′w** had IC_50_ values in the high nanomolar range against *Trypanosoma*, but they showed excellent potency against *L. infantum*. As in the sulfoxide series, toxicity was about ten times lower compared to the parent DAA. Anisyl-substituted sulfones were less potent than their sulfoxide counterparts, with product **4d** even being almost inactive (Table 4). We were surprised by the high insolubility of anisyl-substituted sulfones in almost all types of solvent, even at very low concentrations, contrary to their sulfoxide equivalents. As previously mentioned, the solubility does influence the results of biological assays, whereby the insolubility of compounds **4′d** and **4d** may explain the higher IC_50_ values observed.

#### 2.3.3. Secondary Antileishmanial Evaluation of Diarylideneacetones, 2,6-Diaryltetrahydrothiopyran-4-ones, and Their Related *S*-Oxides

All *S*-oxides and parent DAAs and sulfides were evaluated on the promastigote, the axenic amastigote, and the intra-macrophage amastigote of the wild-type strain of *L. donovani* LV9, as well as the promastigote form of miltefosine-resistant *L. donovani* LV9. Six *S*-oxides showed interesting activities (Appendix A), while compounds with IC_50_ ≥ 100 µM in the promastigote assay were considered inactive and therefore omitted in the amastigote assays.

In addition to the sulfones **4w** and **4′w**, the most active compounds were the sulfoxides derivatives **3** and **3′**. Once again, 2-pyridyl-substituted products were more potent than anisyl derivatives (Appendix A). With regard to their relative sulfides, all the tested *S*-oxides had a higher activity, although the oxidation of the sulfur allows the potency to be restored at a similar level as the parent DAA.

Generally speaking, *S*-oxides are more active on promastigotes (except **3d**) and are also active against intramacrophagic amastigotes, demonstrating that these products may be able to penetrate the macrophage. Considering that most of the *S*-oxides turned out to be toxic against mouse macrophages, another explanation would be that these compounds are damaging the whole macrophage and, at the same time, any *Leishmania* parasites present in these cells. Determining whether these compounds are active by direct interaction with the parasite or by destruction of the host cells will need further investigation, particularly for the new *S*-oxides based on the most active and less toxic substitution patterns recently highlighted in the sulfide series.

#### 2.3.4. Putative Mode-of-Action through a Cascade of Redox Reactions

Sulfide oxidation into sulfoxides or sulfones has been extensively studied in polymer chemistry and medicinal chemistry [57,58,59,60]. In the case of unsaturated Mannich bases [3,25], *S*-oxidation of the dithiol:(bis)Michael adducts can facilitate a β-elimination which reinstates the unsaturated β-ketothioethers (polymer fragmentation) and yields to the release of sulfenic acids (Figure 22.21 in [3]). The latter may be overoxidized in sulfinic and sulfonic acids [57].

Like unsaturated Mannich bases, 4*H*-thiopyran-4-ones might be prone to undergo β–*syn* elimination upon *S*-oxidation and to regenerate DAAs (Figure 7). These DAAs are present as curcuminoids in nature and are prone to react as Michael acceptors with thiols. The DAA reactivity is similar to that of unsaturated Mannich bases described as irreversible inhibitors of large thioredoxin reductases from *Plasmodium falciparum* [3,4a]. These DAAs also irreversibly react with biological dithiols like trypanothione and trypanothione reductase from *T. cruzi* or *T. brucei* [5]. Considering the cascade of redox reactions involving these unsaturated Mannich bases [3], we propose that DAAs enter a cascade of redox reactions in vivo, in concert with the trypanothione regenerating-TR/NADPH system, leading to toxic polymer elongation and sulfur oxide species generation (Figure 7).

In a similar manner, trypanothione reacts with these DAA structures to form oligomers, which were evidenced by ESI-MS analyses [3]. It has been hypothesized that these polymeric entities resulted from a tail-to-tail addition of trypanothione on the electrophilic centers of a DAA molecule. Unlike glutathione, trypanothione has two reactive centers (two thiols) as well as the DAA (two alkenes). Copolymerization of these two monomers was confirmed (Figure 7, step A). It was previously demonstrated that sulfide polymers are prone to be oxidized and undergo polymer fragmentation through β-*syn*-elimination (Figure 7, steps B and C) [57]. Such elimination would regenerate an α,β-unsaturated ketone, which, in turn, could react again with trypanothione, resulting in polymer elongation (Figure 7, step D). In our topic, the repetition of these steps could be considered a redox cycle since a reactive species is sequentially oxidized and reduced, resulting in the futile consumption of trypanothione and NADPH. This redox cycle may finally kill the parasite due to both oxidative stress and the generation of damaging species such as sulfinic or sulfonic acids (Figure 7). Although elusive starting from DAA, such a mechanism has already been described with Mannich bases in polymer chemistry [57] and was therefore considered a plausible rationale for the trypanothione-DAA polymeric entities and the anti-kinetoplastid activity of DAAs. This hypothesis is in accordance with earlier observations made by de Oliveira Silva‘s group, which has reported the oxidative stress generated from potent trypanocidal dibenzylideneacetones against *T. cruzi* by interfering with its redox system [9].

Furthermore, more recent work confirmed that the recovery of Michael-accepting species is driven by oxidation chemistry in nature [61], in which Eelkema’s group demonstrated the redox-controlled reaction cycle under flow conditions for the recovery of Michael acceptors based on thiol-addition and -elimination chemistry. Therefore, we believe that our approach using prodrugs of DAAs mimics nature, where oxidative metabolic processes from 2,6-DA-4-THTPs via their relative *S*-oxides can release DAAs that are highly reactive toward trypanothione upon overoxidation of the thiol-DAA adducts.

## 3. Discussion

We previously demonstrated that the DAA series was not promising for lead development, mainly related to the dramatic cytotoxicity that is likely intrinsic to the structure of these molecules. The prodrug strategy was intended to cope with this issue. Here, we discuss the biological results and check whether this strategy fulfilled the projected study aims. As the library of prodrugs was even smaller than the library of DAAs, establishing a precise correlation between molecular descriptors and anti-kinetoplastid activity was not possible. As such, we only highlight some general trends resulting from the comparison of the activity with several criteria, including the cytotoxicity, stereochemical configuration, and degree of oxidation of the sulfur atom.

### 3.1. Dissociation of Toxicity from Trypanocidal Activity

As previously mentioned, cytotoxicity was the major concern in the DAA series as the most potent DAAs were also the most toxic. We observed that dissociating toxicity from activity would be very difficult. As their reactive electrophilic centers are deactivated, 2,6-DA-4-THTPs and their relative *S*-oxides were expected to be less toxic, as later observed. At first glance for most compounds, we noted some improved selective action against parasites. Considering the potency toward *T. brucei*, a good selectivity was achieved with most compounds, while, toward intramacrophage amastigotes of *T. cruzi* and *L. infantum*/*L. donovani*, mixed results were obtained, with only a few compounds showing selective potency. Despite the latter, the prodrug strategy may be considered promising since the overall toxicity is lowered in this series. In order to obtain a better comprehension of this series, the substitution pattern effects for each stereochemical configuration and oxidation state of the sulfur atom were considered in more detail.

### 3.2. Effects of the Stereochemical Configuration

In the chemistry-related section, we described efforts to develop a novel diastereoselective synthesis of 2,6-DA-4-THTPs. This effort was motivated by the fact that stereochemistry is known to significantly influence biological activities and would therefore have been impossible to deconvolute the results obtained with diastereoisomeric mixtures. Further to the biological results, it was interesting to find out if the configuration of the heterocycle prodrug would really affect the anti-kinetoplastid activity.

To answer this question, we analyzed the results on a complete homogenous series (sulfides and *S*-oxides). As the anisyl substitution did not reveal interesting activities, we focused on the 2-pyridyl series (Table 3). Considering each pair of diastereoisomers, variations were very limited and did not exceed 10% of activity, which can be considered in the error range of the in vitro measurements. Hence, we can conclude that there is no difference in the IC_50_ values between the diastereoisomers of either the sulfides or the *S*-oxides.

Several hypotheses can be suggested to explain the experimental results. The most obvious one is that the activity does not depend on recognizing mechanisms, or at least on stereochemical sensitive ones. Another possibility would be that the *cis* and the (±)-*trans* isomers may have a similar reactivity in a biological medium, resulting in similar activities. Finally, we can hypothesize that, in a biological medium, the (±)-*trans* isomer could be quickly isomerized into the thermodynamically more stable *cis* isomer, which will obviously result in the same activity for both diastereoisomers. The latter is supported by the fact that we observed the isomerization of the (±)-*trans* into the *cis* diastereoisomer in organic conditions (vide supra) and biomimetic conditions (100 µM of product in an aqueous PBS buffer at pH 7.4 for 24 h at 37 °C). Determining which of these three hypotheses is the most plausible will require new experiments and mechanistic studies to determine if an active transport is occurring during compound uptake and to explore the pharmacokinetics in both mammals (e.g., metabolic studies in liver microsomes) and parasites.

### 3.3. Effects of Sulfur Oxidation

Due to their reactivity, which can result in the regeneration of double bonds through β-elimination, *S*-oxides were highlighted in the prodrug strategy as putative metabolites. To assess the real impact of sulfur oxidation, these products were also evaluated in biological assays. As already mentioned, the reactivity of sulfur in β-elimination is higher when the sulfur is oxidized; therefore, we would expect a similar behavior for the anti-kinetoplastid activity. Thus, we compared the antiparasitic activities of sulfide, sulfoxide, and sulfone in two homogenous chemical series, namely the anisyl and 2-pyridyl series (Table 4). Unlike the stereochemical configuration, we noted that the degree of sulfur oxidation significantly affects the anti-kinetoplastid activity. In both series, the oxidation of sulfide to sulfoxide clearly restored the antiparasitic activity that was abolished by the heterocyclic-sulfide formation. In the same way, the over-oxidation of sulfide **2w** to sulfone **4w** resulted in an enhanced activity toward both *T. cruzi* and *T. brucei*. The fact that this was not the case for sulfone **4d**, which was completely inactive, is probably due to solubility issues. All the *S*-oxides were more toxic than their parent sulfides, but this was less important than their gain in activity. In addition, it was well-defined that the 2-pyridyl series showed unacceptable toxicity issues. The same oxidative strategy on more selective substitution patterns (e.g., benzonitrile, toluyl, or *p*-trifluoromethylphenyl) could possibly result in improved activities with lesser toxicity.

## 4. Materials and Methods

### 4.1. Chemistry: General

Commercially available starting materials were purchased from Sigma-Aldrich (Schnelldorf, Germany), ABCR GmbH & Co. KG (Karlsruhe, Germany), Alfa Aesar, and Apollo Scientific (Bredbury, United Kingdom) and were used without further purification. Solvents were obtained from Sigma-Aldrich (Schnelldorf, Germany) and Carlos Erba (Emmendingen, Germany); unless noticed, reagent grade was used for reactions and column of chromatography, and analytical grade was used for recrystallizations. When specified, anhydrous solvents were required; dichloromethane (DCM) was distilled over CaH_2_ under argon. Tetrahydrofuran (THF) was distilled over sodium/benzophenone under argon or dried by passage through an activated alumina column under argon. 1,4-Dioxane and dimethylformamide (DMF) were purchased anhydrous over molecular sieves from Sigma-Aldrich. Triethylamine (Et_3_N), diisopropylethyl amine (DIPEA), pyrrolidine, and piperidine were distilled over KOH under argon and stored over KOH.

All reactions were performed in standard glassware. Microwave reactions were carried out on Biotage Initiator™ or Biotage CEM (Uppsala, Sweden), with comparable results (cross-compared); supplier standard microwave vials were used. Thin Layer Chromatography (TLC) was used to monitor reactions (vide infra).

Crude mixtures were purified, either by recrystallization or by flash column chromatography. The latter was performed using silica gel 60 (230–400 mesh, 0.040–0.063 mm) purchased from E. Merck (Darmstadt, Germany). Automatic flash chromatography was carried out in a Biotage Puriflash apparatus with UV-Vis detection at 254 nm (unless otherwise specified). Monitoring and primary characterization of products were achieved by Thin Layer Chromatography on aluminum sheets coated with silica gel 60 F254 purchased from E. Merck. Eluted TLC was revealed under UV (325 nm and 254 nm) and with chemicals (see composition of TLC dip solutions in Appendix A).

Nuclear Magnetic Resonance (NMR) spectra were recorded on a Bruker AC 300, Bruker AC 400, or Avance DRX500, with solvent peaks as reference. Carbon multiplicities were assigned by Distortionless Enhancement by Polarization Transfer (DEPT) experiments. ^1^H and ^13^C signals were assigned by correlation spectroscopy (COSY), Heteronuclear Single Quantum Correlation (HSQC), and Heteronuclear Multiple-Bond Correlation spectroscopy (HMBC). In the following NMR assignments, coupling constants (*J*) will be expressed in Hertz (Hz), multiplicity is described with (s) as singlet, (d) as doublet, (t) as triplet, and (q) as quadruplet, and a “b-” prefix means that the considered signal is broad. In addition, the following acronyms will be used: ^3^*J_trans_*: ^3^*J* coupling constant between the two alkenyl protons of the enone; ^3^*J*_1,3-TDA_: 1,3-*trans*-diaxial coupling constant between the axial H_X_ proton and the axial H_A_ proton of an ABX system; ^3^*J_cis_*_AE_: *cis*-axial-equatorial coupling constant between the H_X_ proton (either in axial or in equatorial position) and the equatorial H_B_ proton of an ABX system; ArH: aromatic proton; H_vin_: vinylic proton of an enone; C_q_: quaternary carbon; CH_2_: secondary carbon; and Me: methyl group. When a carbon atom is attributed without any doubt to a tertiary aromatic carbon, this carbon is abbreviated to “Ar”. Otherwise, tertiary carbons (aromatic or not) are abbreviated to “CH”. Numbering on structures does not follow the official IUPAC rules; these indications are only provided to make the reading of spectra easier.

Infrared (IR) spectra (cm^−1^) were recorded neat on a Perkin-Elmer Spectrum One Spectrophotometer. UV-Vis spectra were recorded on a Varian Cary 50 spectrophotometer.

ESI-HRMS mass spectra were carried out on a Bruker MicroTOF spectrometer. LC-MS was performed on a ThermoFisher apparatus with ESI ionization. Elemental analyses were obtained from “Service commun d’analyses” from the University of Strasbourg. Melting points (m.p.) were measured on a Stuart Melting Point 10 apparatus and are given uncorrected; when measured after recrystallization, the solvent is mentioned in brackets. For compounds where no elemental analyses were accessible due to their instability, analytical High-Pressure Liquid Chromatography (HPLC) was performed in order to confirm the purity. HPLC experiments were performed on a Hewlett Packard HPLC with dual UV-Vis detection (254 nm and 325 nm) or on a Hitachi HPLC (detection at 254 nm). A standard Nucleosil silica column was used on the first apparatus, while a Nucleodur^®^ C18 was used on the Hitachi apparatus. HPLC retention times (t_R_) were obtained, using the following conditions: method 1–100% eluent A (0.05% trifluoroacetic acid (TFA) in H_2_O) for 5 min, a gradient up to 100% B (0.05% TFA in H_2_O/CH_3_CN 1:4) over 10 min, 100% B for 5 min, followed by a gradient up to 100% A over 5 min at a flow rate of 1 mL/min; method 2–100% eluent A (0.05% trifluoroacetic acid (TFA) in H_2_O) for 5 min, a gradient up to 100% B (0.05% TFA in H_2_O/CH_3_CN 1:4) over 20 min, followed by a gradient up to 100% A over 10 min at a flow rate of 1 mL/min.

In the following sections, solvents will be abbreviated as follows: DCM: dichloromethane; CyHex: Cyclohexane; EtOAc: Ethyl acetate; Tol.: toluene; Et_2_O: diethyl ether; MTBE: methyl tert-butyl ether; and THF: tetrahydrofurane.

A mixture of methanolic ammonia in dichloromethane was used for the elution and purification of highly polar compounds. This was prepared as follows: a commercial solution of 7 M ammonia in methanol (14 mL) was dissolved in methanol (86 mL) and dichloromethane (900 mL). This resulted in a 10% (1 M NH_3_ in methanol) dichloromethane solution, which was then used for TLC and flash chromatography elution. This will be abbreviated as (1 M NH_3_ in MeOH) in DCM.

The synthetic methodologies and analytical data of new symmetrical and unsymmetrical DAAs **1-1′** (procedures **A** to **E**) and the optimized protocols for the synthesis of symmetrical (±)-*trans* and *cis* 2,6-DA-4-THTPs isomers **2-2′** (procedures **F** to **H**), as starting chemicals for the synthesis of their *S*-oxides are described in Appendix A. The new synthetic methodologies and analytical data of 2,6-DA-4-THTPs sulfoxides and sulfones (procedures **I** and **J**, respectively) are presented here. 

### 4.2. Synthesis of 2,6-Diaryl-4H-tetrahydrothiopyran-4-one Sulfoxide and Sulfone Derivatives

#### 4.2.1. Synthesis of “Davis’s Oxaziridine” Reagent

***N*-benzylidenebenzenesulfonamide (5)**Chemical Formula: C_13_H_11_NO_2_SMolecular Weight: 245.30 g·mol^−1^

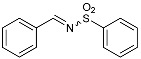



Powdered 4Å molecular sieves (7 g), anhydrous toluene (66 mL), Amberlyst^®^ 15 (120 mg), benzenesulfonamide (6.28 g, 40 mmol, 1.0 equiv.), and benzaldehyde (4.50 mL, 44 mmol, 1.1 equiv.) were introduced in a round-bottomed flask equipped with a Dean-Stark apparatus. The suspension was then heated at 150 °C, under stirring and argon for 17 h. The reaction was then allowed to cool at room temperature and filtered. The solid was washed with toluene (3 × 10 mL), and the liquid phase was evaporated under reduced pressure. The resulting light-yellow oil was crystallized with cold *n*-pentane (if necessary, the flask was stored in the fridge for two hours). Subsequent recrystallization of the white solid in ethyl acetate and pentane afforded the pure desired product **5** as white crystals (7.81 g, 31.8 mmol, 80%). **^1^H NMR (300 MHz, CDCl_3_)** *δ* (ppm): 9.07 (s, 1H, imine), 8.00 (m, 4H), 7.6 (m, 6H). *Fine for the product, according to the literature* [49].
**3-Phenyl-2-(phenylsulfonyl)-1,2-oxaziridine (6), the so-called “Davis’s oxaziridine”**Chemical Formula: C_13_H_11_NO_3_S Molecular Weight: 261.30 g·mol^−1^
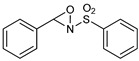


The previously synthesized *N*-benzylidenebenzenesulfonamide **5** (3 g, 12 mmol, 1.0 equiv.) and potassium carbonate (13.8 g, 100 mmol, 8.4 equiv.) were suspended in toluene (60 mL) and water (42 mL) in a two-necked round-bottomed flask cooled at 0 °C. Oxone^®^ (8.9 g, 14.4 mmol, 1.2 equiv.) in water (42 mL) was introduced in a dropping funnel and added dropwise in the flask over 15 min, under stirring, at 0 °C. At the end of the addition, the reaction was allowed to warm to room temperature and was carried on for 45 min. The aqueous phase was separated and extracted with toluene, and the combined organic layers were dried over MgSO_4_ and evaporated under reduced pressure. *Warning*: evaporation must not reach dryness and the rotary bath must be kept under 40 °C. The product was crystallized in the fridge, and the resulting white solid was gently recrystallized in ethyl acetate and *n*-pentane to afford the desired product **6** (2.82 g, 10.8 mmol, 90%) as white crystals; to limit the risk of decomposition, this product was stored at −20 °C under argon. **^1^H NMR (300 MHz, CDCl_3_)** *δ* (ppm): 8.05 (bd d, 2H, Ar), 7.76 (m, 1H, Ar), 7.65 (m, 2H, Ar), 7.44 (m, 5H, Ar), 5.5 (s, 1H, H oxaziridine). **^13^C NMR (75.5 MHz, CDCl_3_)** *δ* (ppm): 135.0 (Ar), 134.7 (C_q_), 131.5 (Ar), 130.5 (C_q_), 129.4 (Ar), 129.3 (Ar), 128.8 (Ar), 128.3 (Ar), 76.3 (CH oxaziridine). *Fine for the product, according to the literature* [49]. 

#### 4.2.2. Synthesis of 2,6-Diaryl-4H-tetrahydrothiopyran-4-one Sulfoxide Derivatives **3**–**3′**

##### General Procedure **I** for the Synthesis of Sulfoxide Derivatives

The sulfide starting material (1.0 equiv.) was dissolved in anhydrous dichloromethane, and the solution was cooled at −20 °C under argon. To this, Davis’s oxaziridine **6** (1.1 equiv.) in anhydrous dichloromethane (final concentration of 0.05 mol·L^−1^) was added. The reaction was carried on for 1h 30 min at −20 °C under argon and then allowed to warm to room temperature for another half hour. The crude product was transferred in a separating funnel with DCM, and the organic layer was washed twice with sodium thiosulfate (0.25 M). The organic layer was dried with anhydrous MgSO_4_ and evaporated to dryness. The resulting residue was purified by column chromatography as mentioned in the following examples.
**(±)-*trans*-2,6-Di-(*p*-anisyl)-*4H*-tetrahydrothiopyran-4-one 1-oxide (3d)**Chemical Formula: C_19_H_20_O_4_SMolecular Weight: 344.42 g·mol^−1^
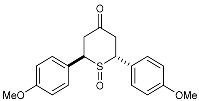


The previously synthesized sulfide **2d** (328 mg, 1 mmol) was used as the starting material and treated according to general procedure **I**. Purification by flash chromatography (SiO_2_, EtOAc/CyHex) gave the desired product **3d** as a white powder (283 mg, 0.82 mmol, 82%). **^1^H NMR (400 MHz, CDCl_3_)** *δ* (ppm): 7.28 (d, ^3^*J* = 8.7 Hz, 2H, H10-H14), 7.07 (d, ^3^*J* = 8.8 Hz, 2H, H15-H19), 6.97 (d, ^3^*J* = 8.8 Hz, 2H, H11-H13), 6.86 (d, ^3^*J* = 8.8 Hz, 2H, H16-H18), 4.78 (dd, ^3^*J* = 6.1 Hz, 2.5 Hz, 1H, H2), 3.85 (s, 3H, OMe), 3.80 (s, 3H, OMe), 3.74 (m, 1H, H_X_), 3.70 (m, 1H, H5_ax_), 3.62 (dd, ^2^*J* = 15.5 Hz, ^3^*J* = 6.1 Hz, 1H, H1_ax_), 3.02 (ddd, ^2^*J* = 15.5 Hz, ^3^*J* = 2.5 Hz, ^4^*J* = 1.7 Hz, 1H, H1_eq_), 2.59 (bdd, ^2^*J* = 12 Hz, ^4^*J* = 1.7 Hz, 1H, H5_eq_). **^13^C NMR (100 MHz, CDCl_3_)** *δ* (ppm): 207.2 (C6), 159.9 (C12 or C17), 159.6 (C12 or C17), 129.3 (C15-19), 129.2 (C10-14), 126.8 (C7 or C8), 125.1 (C7 or C8), 115.0 (C11-C13), 114.4 (C16-C18), 61.8 (C2), 55.5 (OMe), 55.4 (OMe), 54.9 (C4), 38.3 (C5), 35.6 (C1). **IR** (neat) ν (cm^−1^): 1713 (C=O), 1610, 1512, 1255, 1185-1176, 1038 (C-OMe), 959 (S=O), 838-829 (Ar-H). **m.p.** = 173 °C dec. **TLC (SiO_2_):** 70% ethyl acetate in chloroform, R_F_ = 0.27, dark blue with Mostaïne.
***(anti)-* and *(syn)-cis*-2,6-Di-(*p*-anisyl)-*4H*-tetrahydro-thiopyran-4-one 1-oxides (3′d and 3″d)**Chemical Formula: C_19_H_20_O_4_SMolecular Weight: 344.42 g·mol^−1^
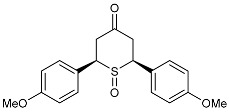


The previously synthesized sulfide **2′d** (328 mg,1 mmol) was used as the starting material and treated according to general procedure **I**. Purification by flash chromatography (SiO_2_, EtOAc/chloroforme) gave the two desired diastereoisomers (Yield: 86%, d.e. 46%): product **3′d** (218 mg, 0.63 mmol, 63%) and product **3″d** (78 mg, 0.23 mmol, 23%) as two white powders. *Major diastereoisomer:*

**^1^H NMR (300 MHz, CDCl_3_)** *δ* (ppm): 7.31 (d, ^3^*J* = 8.7 Hz, 4H, H10), 6.94 (d, ^3^*J* = 8.7 Hz, 4H, H11), 4.07 (AB**X**, ^3^*J*_1,3-TDA_ = 13.8 Hz, ^3^*J_cis_*_AE_ = 2.8 Hz, 2H, H_X_), 3.83 (s, 6H, OMe), 3.73 (**A**BX, ^2^*J*_AB_ = 14.4 Hz, ^3^*J*_1,3-TDA_ = 13.8 Hz, 2H, H_A_), 2.67 (A**B**X, dd, ^2^*J*_AB_ = 14.4 Hz, ^3^*J_cis_*_AE_ = 2.8 Hz, 2H, H_B_). **^13^C NMR (75.5 MHz, DMSO-d6)** *δ* (ppm): 205.4 (C=O), 159.7 (C12), 129.7 (C10), 128.9 (C_q_), 114.7 (C11), 59.9 (SCHx), 55.6 (OMe), 37.9 (CH_2_). **IR** (neat) ν (cm^−1^): 1709 (C=O), 1609, 1513, 1243, 1180, 1028 (C-OMe), 958 (S=O), 837 (Ar-H). **Elemental analysis:** Calcd. C 66.26, H 5.85; Found C 65.96, H 5.66; **m.p.** = 190 °C dec. **TLC (SiO_2_):** 30% ethyl acetate in chloroform, R_F_ = 0.38, purple with Mostaïne.*Minor diastereoisomer:*

**^1^H NMR (300 MHz, CDCl_3_)** *δ* (ppm): 7.29 (d, ^3^*J* = 9.0 Hz, 4H, H10), 6.93 (d, ^3^*J* = 8.8 Hz, 4H, H11), 4.11 (AB**X**, ^3^*J*_1,3-TDA_ = 13.6 Hz, ^3^*J_cis_*_AE_ = 3.0 Hz, 2H, H_X_), 3.82 (s, 6H, OMe), 3.24 (**A**BX, ^2^*J*_AB_ = 15.0 Hz, ^3^*J*_1,3-TDA_ = 13.6 Hz, 2H, H_A_), 3.01 (dd, ^2^*J*_AB_ = 15.0 Hz, ^3^*J_cis_*_AE_ = 3.0 Hz, 2H, H_B_). **^13^C NMR (75.5 MHz, CDCl_3_)** *δ* (ppm): 203.2 (C6), 160.1 (C12), 129.4 (C10), 126.3 (C7), 114.6 (C11), 66.0 (SCHx), 55.3 (OMe), 44.1 (CH2). **IR** (neat) ν (cm^−1^): 1715 (C=O), 1609, 1512, 1253, 1177, 1059 (S=O), 1028 (C-OMe), 839 (Ar-H). **Elemental analysis:** Calcd. C 66.26, H 5.85; Found C 66.08, H 5.76. **m.p.** = 166 °C dec. **TLC (SiO_2_):** 30% ethyl acetate in chloroform, R_F_ = 0.20, purple with Mostaïne.
**(±)-*trans*-2,6-Di-(pyridin-2-yl)-*4H*-tetrahydrothiopyran-4-one 1-oxide (3s)**Chemical Formula: C_15_H_14_N_2_O_2_SMolecular Weight: 286.35 g·mol^−1^
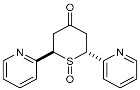


The previously synthesized sulfide **2w** (650 mg, 2.4 mmol) was used as the starting material and treated according to general procedure **I**. Purification by flash chromatography (SiO_2_, MeOH/DCM) gave the desired product **3w** as a white powder (500 mg, 1.75 mmol, 73%). **^1^H NMR (300 MHz, CDCl_3_)** *δ* (ppm): 8.65 (dd, ^3^*J* = 4.8 Hz, ^4^*J* = 1.5 Hz, 2H, H11 or 18), 8.60 (dd, ^3^*J* = 4.8 Hz, ^4^*J* = 1.5 Hz, 2H, H11 or 18), 7.71 (td, ^3^*J* = 7.7 Hz, ^4^*J* = 1.8 Hz, 2H, H13 or 16), 7.72 (td, ^3^*J* = 7.7 Hz, ^4^*J* = 1.8 Hz, 2H, H13 or 16), 7.33 (bd, ^3^*J* = 7.9 Hz, 2H, H14-15), 7.28 (m, 2H, H12-17), 4.73 (t, ^3^*J* = 5.1 Hz, 1H, H2_eq_), 4.56 (dd, ^3^*J*_1,3-TDA_ = 11.1 Hz, ^3^*J_cis_*_AE_ = 3.9 Hz, 1H, H4_ax_), 3.55 (dd, ^2^*J* = 15.8 Hz, ^3^*J*_1,3-TDA_ = 11.1 Hz, 1H, H5_ax_), 3.41 (dd, ^2^*J* = 15.8 Hz, ^3^*J* = 5.1 Hz, 1H, H1_ax_), 3.06 (ddd, ^2^*J* = 15.8 Hz, ^3^*J* = 5.1 Hz, ^4^*J* = 1.3 Hz, 1H, H1_eq_), 2.93 (ddd, ^2^*J* = 15.8 Hz, ^3^*J* = 3.9 Hz, ^4^*J* = 1.3 Hz, 1H, H5_eq_). **^13^C NMR (75.5 MHz, CDCl_3_)** *δ* (ppm): 203.5 (C=O), 154.0 (C7 or C8), 153.2 (C7 or C8), 150.4 (C11 or C18), 149.7 (C11 or C18), 137.2 (C13 or C16), 137.0 (C13 or C16), 124.8 (C14 or C15), 124.9 (C14 or C15), 123.7 (C12 or C17), 123.4 (C12 or C17), 64.3 (C2), 59.7 (C4), 38.0 (C5), 37.0 (C1). **IR** (neat) ν (cm^−1^): 1710 (C=O), 1587 (C=C pyr.), 1471, 1435, 1126, 1037 (S=O), 798-790 (Ar-H). **Elemental analysis:** Calcd. C 62.92, H 4.93, N 9.78; Found C 62.76, 4.83, N 9.66. **m.p.** = 149 °C dec. **TLC (SiO_2_):** 5% methanol in dichloromethane, R_F_ = 0.27, dark blue with Mostaïne.
***(syn/anti)-cis*-2,6-Di-(pyridin-2-yl)-*4H*-tetra-hydrothiopyran-4-one 1-oxide, as diastereoisomeric mixture (3′s)**Chemical Formula: C_15_H_14_N_2_O_2_SMolecular Weight: 286.35 g.mol^−1^
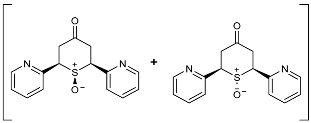


The previously synthesized sulfide **2′w** (500 mg,1.76 mmol) was used as the starting material and treated according to general procedure **I**. Purification by flash chromatography (SiO_2_, MeOH/DCM) gave the desired product **3′w** as a white powder (403 mg, 1.41 mmol, 80%), as a mixture of both diastereoisomers (d.e. 60%). Analyses were performed on the mixture.
*Major diastereoisomer (80 mol%)*

**^1^H NMR (300 MHz, CDCl_3_)** *δ* (ppm): 8.66 (d, ^3^*J* = 5 Hz, 2H, H11), 7.72 (td, ^3^*J* = 7.7 Hz, ^4^*J* = 1.9 Hz, 2H, H13), 7.40 (d, ^3^*J* = 7.7 Hz, 2H, H14), 7.30 (m, 2H, H12), 4.30 (AB**X**, ^3^*J*_1,3-TDA_ = 13 Hz, ^3^*J_cis_*_AE_ = 3.1 Hz, 2H, H_X_), 3.64 (**A**BX, ^2^*J*_AB_ = 16 Hz, ^3^*J*_1,3-TDA_ = 13.1 Hz, 2H, H_A_), 3.02 (A**B**X, ^2^*J*_AB_ = 16 Hz, ^3^*J_cis_*_AE_ = 3.1 Hz, 2H, H_B_). **^13^C NMR (75.5 MHz, CDCl_3_)** *δ* (ppm): 204.6 (C=O), 153.6 (C7), 150.3 (C11), 136.9 (C3), 125.1 (C14), 123.7 (C12), 68.1 (SCHx), 42.1 (CH_2_).
*Minor diastereoisomer (20 mol%)*

**^1^H NMR (300 MHz, CDCl_3_)** *δ* (ppm): 8.66 (d, ^3^*J* = 5 Hz, 2H, H11), 7.75 (td, ^3^*J* = 7.7 Hz, ^4^*J* = 1.6 Hz, 2H, H13), 7.40 (d, ^3^*J* = 7.7 Hz, 2H, H14), 7.30 (m, 2H, H12), 4.48 (AB**X**, ^3^*J*_1,3-TDA_ = 14.1 Hz, ^3^*J_cis_*_AE_ = 3 Hz, 2H, H_X_), 3.83 (dd, ^2^*J* = 14.7 Hz, ^3^*J*_1,3-TDA_ = 14.1 Hz, 2H, H_A_), 2.85 (dd, ^2^*J* = 14.7 Hz, ^3^*J_cis_*_AE_ = 3 Hz, 2H, H_B_). **^13^C NMR (75.5 MHz, CDCl_3_)** *δ* (ppm): 204.8 (C=O), 154.5 (C7), 149.9 (C11), 137.3 (C3), 125.1 (C14), 122.7 (C12), 64.5 (SCHx), 36.41 (CH_2_). **IR** (neat) ν (cm^−1^): 1717 (C=O), 1587 (C=C pyr.), 1466, 1294, 1037/994 (S=O), 835-782 (Ar-H). **Elemental analysis:** Calcd. C 62.92, H 4.93, N 9.78; Found C 62.75, H 5.04, N 9.51. **TLC (SiO_2_):** 10% methanol in dichloromethane, R_F_ = 0.4, dark blue with Mostaïne.

#### 4.2.3. Synthesis of 2,6-Diaryl-4*H*-tetrahydrothiopyran-4-ones Sulfone Derivatives **4**–**4′**

##### General Procedure **J** for the Synthesis of Sulfone Derivatives

The sulfoxide starting material (1.0 equiv.) was solubilized in dichloromethane (final concentration: 0.07 mol·L^−1^). The solution was cooled to 0 °C. When pyridyl-substituted starting materials were used, trifluoroacetic acid (2.0 equiv.) was added to the cold mixture. To this, *m*-chloroperbenzoic acid (1.25 equiv) was added portion-wise. The reaction was carried on overnight, under argon, between 0 °C and 4 °C. The crude was transferred in a separating funnel with DCM, and the organic layer was washed twice with sodium thiosulfate (0.25 M). The organic layer was dried over MgSO_4_ and evaporated to dryness. The resulting residue was purified as mentioned in the following examples.
**(±)-*trans*-2,6-Di-(*p*-anisyl)-*4H*-tetrahydrothiopyran-4-one 1,1-dioxide (4d)**Chemical Formula: C_19_H_20_O_5_SMolecular Weight: 360.42 g·mol^−1^
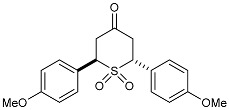


The previously synthesized sulfoxide **3d** (325 mg, 0.94 mmol) was used as the starting material and treated according to general procedure **J**. Purification by flash chromatography (SiO_2_, EtOAc/CyHex) gave the desired product **4d** as a white powder (262 mg, 0.72 mmol, 77%). **^1^H NMR (300 MHz, CDCl_3_)** *δ* (ppm): 7.32 (d, ^3^*J* = 8.8 Hz, 4H, H10), 6.94 (d, ^3^*J* = 8.8 Hz, 4H, H11), 4.37 (AB**X**, ^3^*J*_AX_ = 8.2 Hz, ^3^*J*_BX_ = 5.0 Hz, 2H, H_X_), 3.83 (s, 6H, OMe), 3.42 (**AB**X, ^2^*J*_AB_ = 16 Hz, ^3^*J*_AX_ = 8.2 Hz, ^3^*J*_BX_ = 5.0 Hz, Δν = 42 Hz, 4H, H_A_H_B_). **^13^C NMR (75.5 MHz, CDCl_3_)** *δ* (ppm): 203.9 (C=O), 160.5 (C12), 130.7 (C10), 122.0 (C7), 114.5 (C11), 60.5 (SCHx), 55.3 (OMe), 44.9 (CH_2_). **IR** (neat) ν (cm^−1^): 1715 (C=O), 1608, 1299, 1252, 1120 (-SO_2_), 1029-1021 (C-OMe), 837 (Ar-H). **Elemental analysis:** Calcd. C 63.32, H 5.59; Found C 63.19, H 5.61. **m.p.** = 160 °C. **TLC (SiO_2_):** 70% ethyl acetate in cyclohexane, R_F_ = 0.54, dark blue with Mostaïne.
***cis*-2,6-Di-(*p*-anisyl)-*4H*-tetrahydrothiopyran-4-one 1,1-dioxide (4′d)**Chemical Formula: C_19_H_20_O_5_SMolecular Weight: 360.42 g.mol^−1^
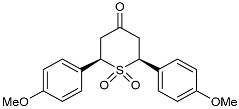


The previously synthesized sulfoxide (mixture of **3′d** and **3″d** without separation) (344 mg, 1 mmol) was used as the starting material and treated according to general procedure **J**. Purification by flash chromatography (SiO_2_, EtOAc/CyHex) gave the desired product **4′d** as a white powder (288 mg, 0.8 mmol, 80%). **m.p.** = 190–192 °C. **^1^H NMR (300 MHz, CDCl_3_)** *δ* (ppm): 7.35 (d, ^3^*J* = 9.1 Hz, 4H, H10-H14), 6.93 (d, ^3^*J* = 9.1 Hz, 4H, H11-H13), 4.47 (AB**X**, ^3^*J*_1,3-TDA_ = 14.2 Hz, ^3^*J_cis_*_AE_ = 2.4 Hz, 2H, H_X_), 3.83 (s, 6H, OMe), 3.42 (**A**BX, bt, *J* ≈ 15 Hz, H_A_), 2.93 (A**B**X, bd, *J* ≈ 15 Hz, H_B_). **^13^C NMR (75.5 MHz, CDCl_3_)** *δ* (ppm): 202.5 (C=O), 160.9 (C12), 130.9 (C10), 122.5 (C7), 114.7 (C11), 64.0 (SCHx), 55.5 (OMe), 46.2 (CH_2_). **TLC (SiO_2_):** 50% ethyl acetate in cyclohexane, R_F_ = 0.22, purple with Mostaïne.
**(±)-*trans*-2,6-Di-(pyridin-2-yl)-4*H*-tetrahydrothiopyran-4-one 1,1-dioxide (4w)**Chemical Formula: C_15_H_14_N_2_O_3_SMolecular Weight: 302.35 g.mol^−1^
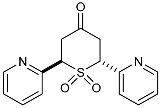


The previously synthesized sulfoxide **3w** (315 mg, 1.1 mmol) was used as the starting material and treated according to general procedure **J**. Purification by recrystallization in a boiling mixture of methanol and chloroform (1:1) gave the desired product **4w** as a white powder (120 mg, 0.4 mmol, 36%). **^1^H NMR (300 MHz, DMSO-*d_6_*)** *δ* (ppm): 8.61 (dd, ^3^*J* = 4.9 Hz, ^4^*J* = 1.8 Hz, 2H, H11), 7.87 (td, ^3^*J* = 7.7 Hz, ^4^*J* = 1.8 Hz, 2H, H13), 7.50 (bd, ^3^*J* = 7.7 Hz, 2H, H14), 7.47 (ddd, ^3^*J* = 7.7 Hz, 4.9 Hz, ^4^*J* = 1.0 Hz, 2H, H12), 5.23 (AB**X**, ^3^*J* = 7.8 Hz, 5.1 Hz, 2H, H_X_), 3.4-3.3 (**A**BX, m, 2H, H_A_), 3.22 (A**B**X, dd, ^2^*J*_AB_ = 16.1 Hz, ^3^*J*_BX_ = 5.1 Hz, 2H, H_B_). **^13^C NMR (75.5 MHz, DMSO-*d_6_*)** *δ* (ppm): 201.5 (C=O), 150.9 (C7), 149.4 (C11), 137.2 (C13), 125.9 (C14), 124.1 (C12), 63.3 (SCHx), 43.2 (CH2). **IR** (neat) ν (cm^−1^): 1716 (C=O), 1589 (C=C pyr.), 1471, 1119 (-SO_2_), 838-750 (Ar-H). **HRMS (ESI+):** [M + Na]^+^ Calcd. *m*/*z* 325.062; Found m/z 325.062. **m.p.** = 207 °C dec. (MeOH/Chloroform). **TLC (SiO_2_):** 5% methanol in dichloromethane, R_F_ = 0.55, orange with Dragendorff.
***cis*-2,6-Di-(pyridin-2-yl)-*4H*-tetrahydrothiopyran-4-one 1,1-dioxide (4′w)**Chemical Formula: C_15_H_14_N_2_O_3_SMolecular Weight: 302.35 g.mol^−1^
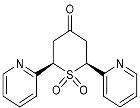


The previously synthesized diastereoisomeric mixture of sulfoxide **3′w** (236 mg, 0.82 mmol) was used as the starting material and treated according to general procedure **J**. Purification by recrystallization in boiling chloroform gave the desired product **4′w** as a white powder (230 mg, 0.75 mmol, 92%). **^1^H NMR (300 MHz, DMSO-*d_6_*)** *δ* (ppm): 8.68 (dd, ^3^*J* = 4.8 Hz, ^4^*J* = 1.8 Hz, 2H, H11), 7.76 (td, ^3^*J* = 7.8 Hz, ^4^*J* = 1.8 Hz, 2H, H13), 7.50 (bd, ^3^*J* = 7.8 Hz, 2H, H14), 7.47 (ddd, ^3^*J* = 7.8 Hz, 4.8 Hz, ^4^*J* = 1.0 Hz, 2H, H12), 4.93 (AB**X**, ^3^*J*_1,3-TDA_ = 13.0 Hz, ^3^*J_cis_*_AE_ = 3.8 Hz, 2H, H_X_), 4.05 (**A**BX, ^2^*J*_AB_ = 15.5 Hz, ^3^*J*_1,3-TDA_ = 13.0 Hz,, 2H, H_A_), 3.09 (A**B**X, ^2^*J*_AB_ = 15.5 Hz, ^3^*J_cis_*_AE_ = 3.8 Hz, 2H, H_B_). **^13^C NMR (75.5 MHz, DMSO-*d_6_*)** *δ* (ppm): 201.5 (C=O), 150.1 (C11), 148.2 (C7), 136.9 (C13), 125.5 (C14),124.3 (C12), 66.2 (SCHx), 43.8 (CH2). **HRMS (ESI+):** [M+Na]^+^ Calcd. m/z 325.062; Found *m*/*z* 325.061. **m.p.** = 210 °C dec. (Chloroform). **TLC (SiO_2_):** 10% methanol in dichloromethane, R_F_ = 0.74, orange with Dragendorff.

### 4.3. Biological Assays

#### 4.3.1. Assays from the University of Paris-Saclay (Prof. P. Loiseau, BioCIS, UMR 8076 CNRS)

##### *L. donovani* (MHOM/ET/67/HU3)—Promastigote Stage

*L. donovani* promastigote forms from the logarithmic phase culture were suspended to yield 10^6^ cells/mL. Miltefosine was used as the antileishmanial reference compound. Compounds to be evaluated and miltefosine were distributed in the plates by making a serial dilution. The final concentrations used were between 100 µM and 50 nM. Triplicates were used for each concentration. After a 3-day incubation period at 27 °C in the dark and under a 5% CO_2_ atmosphere, the viability of the promastigotes was assessed using the tetrazolium-dye (MTT) colorimetric method, which measures the reduction of a tetrazolium component (MTT) into an insoluble formazan product by the mitochondria of viable cells. After the incubation of the cells with the MTT reagent, a detergent solution (Triton X100, HCl) was added to lyse the cells and dissolve the colored crystals. The absorbance at 570 nm, directly proportional to the number of viable cells, was measured using an ELISA plate reader. The results were expressed as the concentrations inhibiting parasite growth by 50% (IC_50_) ± SD after a 3-day incubation period.

##### *L. donovani* (MHOM/ET/67/HU3)—Amastigote Stage

*L. donovani* amastigote forms were suspended to yield 10^7^ cells/mL. The final compound concentrations used were 100 µM, 10 µM, 1 µM, 500 nM, and 50 nM. Duplicates were used for each concentration. Cultures were incubated at 37 °C for 72 h in the dark and under a 5% CO_2_ atmosphere; then, the viability of the amastigotes was assessed using the SYBR^®^ green I (Invitrogen, France) incorporation method. Parasite growth was determined by using SYBR^®^ Green I, a dye with marked fluorescence enhancement upon contact with parasite DNA. Parasites were lysed following Direct PCReCell Genotyping without DNA isolation protocol (Euromedex, Souffelweyersheim, France). A total of 10 mL of SYBR Green I was added to each well, and the contents were mixed. Fluorescence was measured with Mastercylcer^®^ ep realplex (Eppendorf, Montesson, France). The fluorescence obtained was compared to those from the range obtained with different parasite densities.

#### 4.3.2. Assays from the University of Antwerp (Prof. L. Maes)

##### *Trypanosoma brucei brucei* 

A suramin-sensitive strain (Squib 427) was incubated in 96-well plates at five concentrations of the compound (64, 16, 4, 1, and 0.25 μM). After 3 days, parasite growth was assessed fluorimetrically following the addition of resazurin. After 24 h at 37 °C, fluorescence was measured (λ_ex_ = 550 nm, λ_em_ = 590 nm). The results were expressed as a percentage reduction in parasite growth/viability compared to control wells, and a 50% inhibitory concentration (IC_50_) was calculated. Suramin was included as a reference drug (IC_50_ = 0.12 ± 0.07 μM).

##### *Trypanosoma cruzi* 

A nifurtimox-sensitive strain (Tulahuen CL2 β-galactosidase) in human lung fibroblast *h*MRC-5_SV2_ cells was incubated for 7 days in 96-well plates with five concentrations of compound (64, 16, 4, 1, and 0.25 μM). The parasitic burden was evaluated through the activity of the enzyme β-galactosidase on a specific chlorophenol red ß-D-galactopyranoside (CPRG) substrate. Live parasites converted the yellow-orange CPRG substrate into the red chromophore chlorophenol red, yielding a dark-red solution, which could be quantitatively measured at 540 nm after 4 h of incubation at 37 °C. The results were expressed as a percentage reduction in parasite growth/viability compared to control wells, and an IC_50_ was calculated. Nifurtimox was included as a reference drug (IC_50_ = 0.845 ± 0.2 μM).

##### *Leishmania spp.* 

*L. infantum* (MHOM/MA(BE)/67 strain) amastigote were incubated for 5 days in 96-well plates with five concentrations of compound (64, 16, 4, 1, and 0.25 μM). Primary peritoneal mouse macrophages were used as host cells. Parasite burdens (i.e., the number of amastigotes per macrophage) were microscopically assessed. The results were expressed as a percentage reduction in parasite burden compared to untreated control wells, and an IC_50_ was calculated. Pentostam^®^ (IC_50_ = 6.8 ± 0.9 μM) and miltefosine (IC_50_ = 5.2 ± 0.8 μM) were included as reference drugs.

##### Toxicity 

The cytotoxicity of compounds was evaluated toward human diploid embryonic lung fibroblast *h*MRC-5_SV2_ cells, derived from fetal lung tissue and first described in 1970. *h*MRC-5_SV2_ cells were cultured in 96-well plates with five concentrations of the compound (64, 16, 4, 1, and 0.25 μM). After 3 days of incubation, the cell viability was assessed fluorimetrically after the addition of resazurin, and fluorescence was measured (λ_ex_ = 550 nm, λ_em_ = 590 nm). The results were expressed as a percentage reduction in cell growth/viability compared to control wells, and a 50% cytotoxic concentration (CC_50_) was calculated. The reference cytotoxic drugs were tamoxifen or ivermectin. Additionally, toxicity toward mouse macrophages was qualitatively observed. Three concentrations were included (64, 16, and 4 µM), and the range of toxicity was ranked from the least toxic to the most toxic compound.

## 5. Conclusions

In spite of a potent anti-kinetoplastid activity, previously reported diarylideneacetones suffered from a recurrent toxicity issue that seems intrinsically linked to the structure of this series. As the highly reactive enone system was hypothesized to be responsible for this host toxicity, we synthesized diarylideneacetone prodrugs with the goal of designing a new structure where the enone system is temporarily masked in a heterocyclic structure. Enabling the regeneration of double bonds of diarylideneacetones through β-elimination, the 2,6-DA-4-THTP scaffold was selected for this prodrug strategy. We hypothesized that relative *S*-oxide derivatives might also be putative active metabolites due to their higher reactivity compared to sulfides.

With the constant aim of developing mild reaction conditions that would be compatible with all the substitution patterns required by this medicinal chemistry project, novel synthetic protocols were developed. Building upon our seminal work on the chemistry of 2,6-DA-4-THTPs, we used two approaches to selectively synthesize the *cis* and (±)-*trans* isomers. A subset of the resulting library of sulfide was used to produce the corresponding *S*-oxides. Sulfoxide derivatives were obtained through the use of Davis’s oxaziridine, which proved to be highly chemoselective. Sulfone derivatives were synthesized by the slow and mild oxidation of the sulfoxide with m-chloroperbenzoic acid. The synthetic work performed on this new series allowed the selective synthesis of each diastereoisomer on demand (as summarized in Appendix A). Further to the optimization work, each step was controlled and performed under mild conditions, which are compatible with a wide panel of substitution patterns.

Both libraries of sulfides and *S*-oxides were evaluated for selective anti-kinetoplastid activity potential in biological assays against *T. cruzi*, *T. b. brucei*, *L. donovani*, and *L. infantum*. Generally speaking, most of the sulfides were not toxic to human cells but showed only moderate or no antiparasitic activity. Nevertheless, several substitution patterns were identified as promising scaffolds: *p*-phenyl-, *p*-benzonitrile-, *p*-trifluoromethylphenyl-, or *p*-tolyl-substituted 2,6-DA-4-THTPs. We noted that the heterocycle stereochemistry seemed to affect neither the antiparasitic activity nor the cytotoxicity. We could finally demonstrate that *S*-oxide putative metabolites showed an enhanced potency toward parasites, while the toxicity toward human cells remained limited. Hence, we may consider that the first objective of the prodrug strategy has been achieved as the heterocycle formation resulted in a substantial loss of cytotoxicity. However, this loss is counterbalanced by a concomitant reduction in anti-kinetoplastid potency. Preliminary results obtained with the *S*-oxides series further suggested that an oxidation of the sulfur atom, either in chemical or biological conditions, is liable to allow a partial recovery of selective activity, but this must be confirmed by further experiments, especially with *S*-oxides substituted with the most active patterns.

Despite its great use potential, 2,6-DA-4-THTPs have not been extensively described for drug discovery. Our present work aims at filling this gap. We discovered and optimized two approaches to selectively synthesize the two isomers of 2,6-DA-4-THTPs, namely the *cis* and (±)-*trans*. These procedures were developed as safe, reliable, and efficient alternatives to the previously published protocols, especially through the substitution of toxic gaseous hydrogen sulfide by sodium sulfide hydrate. This methodology was applied to multigram synthesis, with good to excellent yields, both with electron donating or withdrawing substitutions and with satisfactory diastereoisomeric excess. The spectroscopic data correlated and allowed fast and reliable determination of the exact product configuration. As sulfoxide derivatives of 2,6-DA-4-THTPs have not yet been researched, we proposed a new synthesis based on the use of the highly chemoselective, safe, and non-toxic Davis’s oxaziridine. The mild oxidation conditions proved to be compatible with a wide range of substrates, including heterocyclic-sensitive ones. Finally, we demonstrated that sulfone derivatives could be synthesized in excellent yields by the smooth and slow oxidation of parent sulfoxides with mCPBA.

To summarize, we demonstrated that the oxidation of the sulfur atom significantly improves the activity potential toward kinetoplastid parasites. The residual toxicity issues could possibly be solved by the use of even more selective substitution patterns. Based on our preliminary results, sulfoxide derivatives are more promising as they are more soluble than their relative sulfides, parent DAA, or even their sulfone counterparts. The observations on the effect of the oxidation need further substantiation with several additional new substitution patterns.

## 6. Patent

Wenzel, I. N.; Müller, T. J. J.; Hanquet, G.; Lanfranchi, D. A.; Leroux, F.; Gendron, T.; Davioud-Charvet, E. Dibenzylidene- and heteroarylideneacetone derivatives as kinetoplastideae parasiticides and their preparation, pharmaceutical compositions, and use in the treatment of trypanosomiasis and leishmaniasis. EP 2009-290719.5—2103 (CNRS, Strasbourg University, Heidelberg University, 18 September 2009). Extension PCT Int. Appl. PCT/EP2010/063825 (20 September 2010), WO 2011033115 A2 20110324 (2011). 

## Data Availability

Samples of the compound **1**–**4** series are available from the authors.

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
