# Peer review of "Chemoselective Synthesis and Anti-Kinetoplastidal Properties of 2,6-Diaryl-4H-tetrahydro-thiopyran-4-one S-Oxides: Their Interplay in a Cascade of Redox Reactions from Diarylideneacetones"

_molecules, 2024, doi:10.3390/molecules29071620_

Round 1

Reviewer 1 Report

Comments and Suggestions for Authors

The article prepared by Thibault Gendron et al. presented a study about synthesis and anti-kinetoplastidal properties of 2,6- diaryl-4H-tetrahydro-thiopyran-4-one S-oxides derivatives. The topic is interesting due to biological aspects as well as future application of these compounds in medicine. The Authors deserves appreciation for carrying out this captivating research.

I propose to publish article after major corrections. Kindly consider the suggestions that have been offered:

(1)   Obligatory, the Authors must use a template and refresh the manuscript.

(2)   Please add information into the article which of the synthesized compounds are novel. Is it just the technique that is new? This should be clearly underlined (abstract). A similar information should be included in the main text.

(3)   The sentence "The observed cytotoxicity is likely due to high reactivity of the electrophilic centers of DAA with human glutathione. " needs confirmation in the literature. Please add information about a similar study.

(4)   Scheme 1 – the product does not have a number. Is it correct?

(5)   Scheme 2 requires revision. Why do the diagrams show compounds 2a-2za, which ultimately form 3 and 3’? Explain "Ar". Obligatory, the numbering system must be created again.

(6)   I am not sure whether the experimental details included in the "Scheme 2" description are necessary. It is better to move it as a discussion into the main text.

(7)   Please review section 2.1 again. Compounds 1' are included in the title, but no have information about their in the text below.

(8)   Section 2.2 – Please add number for “2,6-diaryl-4H-tetrahydrothiopyran-4-one S-oxides”. Apply this to the entire manuscript.

(9)   Figure 3 – please check the conformation.

(10) Scheme 4 – “Synthesis of cis and (±)-trans 2,6-DA-4-THTP sulfone derivatives 4 and 4’” but information only about a selected connection.

(11) For each compound (or series), please add a method of monitoring the reaction, including the type of solvent and other details necessary to reproduce the synthesis.

(12) In the supporting information section, please add spectra for new compounds. For compounds that are known, please add literature references and show that the spectral data is consistent with the literature.

(13) Sometimes the text in the manuscript is chaotic. I suggest the Authors revise the whole article and change the discussion for better order.

(14) Please consider removing the "Abbreviations" section. There are not many of them. Explaining them in the main text is enough.

(15) Please use the proper numbering system for Molecules journal in the references section.

Reviewer 2 Report

Comments and Suggestions for Authors

Elisabeth Davioud-Charvet and coworkers reported the synthesis of 2,6-diaryl-4H-tetrahydro-thiopyran-4-one S-oxides and their evaluation of the anti-kinetoplastidal properties. They used a prodrug strategy to lower the major toxicity. According to the result of the evaluation, the S-oxides can be successfully employed for the substitution of the previous Michael acceptors. The masked Michael acceptors are less toxic against the mammal cells. Moreover, the explanation of the masked drugs is reasonable. It’s a high-quality paper, the author solved the problem of the side effects of the previous drugs, showed their prodrug strategy and gave reasonable explanation.

Minor things:

1.     In the introduction part, the author should draw their design, so that the readers can understand the paper more clearly

2.     In scheme2, condition C, the d.e. value should be provided.

3.     The authors have synthesized a series of S-oxides with different substitution. The substitution effect should be discussed.

In summary, this paper can be accepted after minor revision

Round 2

Reviewer 1 Report

Comments and Suggestions for Authors

After a thorough review of the submitted version, my decision is to accept it in its current form.